

# An examination of the impact of Olson's extinction on tetrapods from Texas

Neil Brocklehurst

Museum für Naturkunde, Leibniz-Institut für Evolutions- und Biodiversitätsforschung, Berlin, Germany

## ABSTRACT

It has been suggested that a transition between a pelycosaurian-grade synapsid dominated fauna of the Cisuralian (early Permian) and the therapsid dominated fauna of the Guadalupian (middle Permian) was accompanied by, and possibly driven by, a mass extinction dubbed Olson's Extinction. However, this interpretation of the record has recently been criticised as being a result of inappropriate time-binning strategies: calculating species richness within international stages or substages combines extinctions occurring throughout the late Kungurian stage into a single event. To address this criticism, I examine the best record available for the time of the extinction, the tetrapod-bearing formations of Texas, at a finer stratigraphic scale than those previously employed. Species richness is calculated using four different time-binning schemes: the traditional Land Vertebrate Faunachrons (LVFs); a re-definition of the LVFs using constrained cluster analysis; individual formations treated as time bins; and a stochastic approach assigning specimens to half-million-year bins. Diversity is calculated at the genus and species level, both with and without subsampling, and extinction rates are also inferred. Under all time-binning schemes, both at the genus and species level, a substantial drop in diversity occurs during the Redtankian LVF. Extinction rates are raised above background rates throughout this time, but the biggest peak occurs in the Choza Formation (uppermost Redtankian), coinciding with the disappearance from the fossil record of several of amphibian clades. This study, carried out at a finer stratigraphic scale than previous examinations, indicates that Olson's Extinction is not an artefact of the method used to bin data by time in previous analyses.

## INTRODUCTION

A faunal turnover of tetrapods has long been recognised between the Cisuralian and Guadalupian (early and middle Permian, respectively). The former is characterised by a diverse array of amphibians, pelycosaurian-grade synapsids (particularly carnivorous sphenacodontids and herbivorous edaphosaurids), and captorhinids, whereas the latter is dominated by therapsid synapsids, with increased diversity of parareptiles and amphibian diversity substantially reduced (*Olson, 1962*; *Olson, 1966*; *Kemp, 2006*; *Sahney & Benton, 2008*; *Ruta et al., 2011*; *Benton, 2012*; *Benson & Upchurch, 2013*; *Brocklehurst, Kammerer & Fröbisch, 2013*; *Brocklehurst et al., 2017*). These faunal changes were accompanied by ecological shifts, including a transition towards more complex ecosystems with abundant

Corresponding author
Neil Brocklehurst,
neil.brocklehurst@mfn-berlin.de

large tetrapods as primary consumers (*Olson, 1966*). However, the nature and progress of the transition is still strongly debated.

The possibility of a mass extinction accompanying this transition was first suggested by *Olson (1982)*, who noted a drop in the number of families across Cisuralian/Guadalupian boundary. This drop was principally concentrated among amphibian families (amniote diversity was shown to increase slightly). *Sahney & Benton (2008)* provided a more detailed examination of diversity through the Permian, still at the family level but with temporal resolution at the stage level. Decreases in both species richness, diversification rate and ecological diversity were apparent through the Kungurian and Roadian (the last stage of the Cisuralian and the first of the Guadalupian respectively). *Sahney & Benton (2008)* dubbed this mass extinction event "Olson's Extinction" and hypothesised that it might have been a causal factor in the faunal turnover occurring at this time.

The hypothesis of Sahney & Benton has been criticised as being based on family-level data that was not corrected for sampling heterogeneity (*Benson & Upchurch, 2013*; *Brocklehurst, Kammerer & Fröbisch, 2013*). Nevertheless, subsequent studies both of tetrapods as a whole (*Benton, 2012*; *Benton et al., 2013*; *Benson & Upchurch, 2013*; *Brocklehurst et al., 2017*) and subgroups within Tetrapoda (*Ruta & Benton, 2008*; *Ruta et al., 2011*; *Brocklehurst, Kammerer & Fröbisch, 2013*; *Brocklehurst et al., 2015*), carried out at species and genus levels and employing a variety of sampling correction methods, have identified diversity decreases across the Kungurian/Roadian boundary.

Despite this, the theory of Olson's Extinction has been criticised in other ways. *Benson & Upchurch (2013)* suggested that the mass extinction was an artefact of the geographically patchy fossil record. The record from the Cisuralian is known almost entirely from palaeoequatorial localities, particularly from North America and Europe, whereas that of the Guadalupian is dominated by palaeotemperate localities from Russia and South Africa (*Lucas, 2004*; *Kemp, 2006*). Not only does this make it difficult to ascertain over what timescale the extinction took place and to what extent the transition was a global event, but the apparent diversity drop might simply represent a latitudinal diversity gradient (*Benson & Upchurch, 2013*). In most modern clades, diversity is higher in equatorial regions than temperate regions (*Willig, Kaufman & Stevens, 2003*; *Hillebrand, 2004*), and so it was argued that the shift in sampling locality from more diverse to less diverse latitudes might be the cause of the apparent decrease in species richness (*Benson & Upchurch, 2013*). *Brocklehurst et al. (2017)*, however, argued against this point of view. It has been noted that the latitudinal diversity gradient was not a constant feature through geological time (*Archibald et al., 2010*; *Rose et al., 2011*; *Yasuhara et al., 2012*; *Mannion et al., 2012*; *Mannion et al., 2014*), and it was demonstrated that, in the few Permian time bins where tetrapod data was available from both palaeoequatorial and palaeotemperate latitudes, the temperate latitudes exhibited higher species richness after correcting for sampling (*Brocklehurst et al., 2017*).

Further criticism of Olson's Extinction was put forward by *Lucas (2017a)*. Lucas argued that the inference of a mass extinction across the Kungurian/Roadian at this time was an artefact of two confounding factors. First, the majority of the studies cited used geological

stage or substages as their time bins, thus conflating extinctions occurring throughout the Kungurian into a single event.

Second, *Lucas (2017a)* argued that incorrect ages were applied to numerous geological formations, in particular the San Angelo and Chickasha formations of Texas and Oklahoma, respectively. The ages of these formations have long been a point of contention. Early estimates placed them in the latest Leonardian (late Kungurian) (*Lucas & Heckert, 2001*; *Lucas, 2004*), but discovery of a specimen, from Chickasha, of the parareptile *Macroleter*, previously only known from the Middle Permian of Russia, caused *Reisz & Laurin (2001)* to argue for an equivalency between this formation and the Kazanian-aged (earliest Guadalupian) faunas of Russia. *Lucas (2002)* rejected their arguments based on the ammonite fauna of the Blaine Formation, a marine formation immediately overlying the San Angelo, which he claimed supported a Leonardian age. *Reisz & Laurin (2002)* criticised the interpretation of Lucas, suggesting that a key taxon in the arguments had a much longer range than suggested and highlighting previous studies of the Blaine formation interpreting it as Guadalupian in age. *Lozovsky (2003)* also used ammonite biostratigraphy to support a Roadian age for the Chickasha and San Angelo formations, and these ages have been adopted in most subsequent studies (e.g., *Sahney & Benton, 2008*; *Benton, 2012*; *Brocklehurst, Kammerer & Fröbisch, 2013*; *Brocklehurst et al., 2017*). However, *Lucas (2017a)* still supports a latest Kungurian age for these two formations. He therefore suggested that an extinction across the Kungurian/Roadian boundary cannot be assessed in a global framework, as there is no stratigraphic overlap between the North American and Russian formations.

It is not the purpose of this paper to argue against these two criticisms of *Lucas (2017a)*. Indeed, I am fully prepared to agree that time-binning strategies employing the geological stages or substages, while often necessary for global analyses where the correlations between the regional biostratigraphic schemes are inexact, have the potential to produce spurious results. Such binning strategies produce time-averaged diversity estimates for a time bin that can differ from the true standing diversity at any one time in the bin (*Raup, 1972*; *Lucas, 1994*; *Foote, 1994*; *Miller & Foote, 1996*; *Alroy, 2010a*; *Gibert & Escarguel, 2017*). Instead it is my intention to approach the question of Olson's extinction from a different angle, one that addresses the issues of binning strategy while bypassing the disagreements surrounding the ages of the San Angelo and Chickasha formations. In fact, the framework of this analysis is one suggested by *Lucas (2017a)* himself: when the fossil record is geographically patchy with uncertain global correlations, it is better to study mass extinctions using the "best sections" method, focussing one or a few well sampled, stratigraphically dense fossiliferous sections to examine the progress of the extinction. While only providing a local perspective on the event under study, this method does allow more detailed analysis than is provided in global studies with coarse temporal resolution.

The "best section" of tetrapods in the Cisuralian is doubtless that of Texas, which represents a reasonably continuous sequence from the late Carboniferous until the end of the Cisuralian (*Romer, 1928*; *Romer, 1935*; *Hook, 1989*; *Lucas, 2006*; *Lucas, 2017a*). A detailed examination of the Cisuralian tetrapod record from Texas, covering the stratigraphic sequence from the Pueblo Formation until the San Angelo Formation,

allows much higher resolution than previous studies. Moreover, it renders the debate regarding the age of the San Angelo formation moot. The issue is no longer whether there is a Kungurian/Roadian boundary event, but instead whether an extinction event is identified between the Redtankian and Littlecrotonian land vertebrate faunachrons (biostratigraphic time bins based on the tetrapod fossil record, the former correlating in Texas with the Clear Fork Group, the latter with the San Angelo Formation). The presence of an extinction event between these two faunachrons is assessed at both genus and species levels, with four different time-binning systems and results shown both with and without sampling correction.

## MATERIALS AND METHODS

### Data

Data on the number of specimens of tetrapod species in each time bin was assembled from a variety of sources. The primary literature and the paleobiology database, downloaded from the fossilworks website (http://fossilworks.org) on October 2017, were the principal sources, but were supplemented by observation of specimens in museum collections and also by data sent from some museums (Museum of Comparative Zoology, Harvard; Field Museum of Natural History, Chicago; American Museum of Natural History, New York; Yale Peabody Museum, New Haven; University of California Museum of Palaeontology, Berkeley; Sam Noble Oklahoma Museum of Natural History, Norman). Specimens of uncertain provenance were not included. The data was examined at both species and genus level. While it has often been the preference to examine data at the species-level, *Sepkoski (1984)* argued that as the species are the real "units" of evolution, it is at that level that evolution should be studied), *Lucas (2017a)* suggested that the genus is preferable for early Permian tetrapods to avoid the influence of large numbers of singletons (single-specimen taxa), which under poor sampling produce a great deal of "noise" in the evolutionary signal (*Alroy, 1998*; *Foote, 2000*). The data does not include a large number of species represented by only a single specimen (18 out of 102), but more than half (65) the taxa represent single-occurrences (present in only one formation). The final datasets are provided in Data S1 and S2.

### Time bins

Four methods were used to define time bins, each successively dividing the early Permian into smaller portions of time. The first set of bins used are the land vertebrate faunachrons (LVFs): the biostratigraphic bins based on the first and last appearances of key tetrapod genera (*Lucas, 1998*). As these are biostratigraphic bins, their boundaries should correspond to major periods of turnover among tetrapods, and so the diversity estimates within each faunachron should provide a better approximation of the standing diversity at any point in time than using the international stages. In fact, since the boundaries of the Cisuralian LVFs are primarily based on the section under study, they are more likely to coincide with events relevant to the taxa under study.

The second binning scheme used represents a redefinition of the land vertebrate faunochrons using a clustering approach. CONISS is a constrained clustering analysis,

which groups stratigraphic sections into hierarchical clusters based on the taxonomic distances between, while maintaining the order of the stratigraphic sequence (*Grimm, 1987*). The taxonomic distances between the formations were calculated using *Alroy*'s (*2015a*) modification of the Forbes metric, applying the RAC correction suggested by *Brocklehurst, Day & Fröbisch (2018a)* to account for differences in the evenness of the relative abundance distributions, which with incomplete sampling can bias the distances observed. The functions to carry out the RAC method are available on Dryad (*Brocklehurst, Day & Fröbisch, 2018b*). The CONISS analysis was carried out in R version 3.3.2 (*R Core Team, 2017*), using functions from the package rioja (*Juggins, 2009*). The boundaries of the original LVFs were then shifted to ensure that formations which were clustered together were grouped in the same bin.

The third binning scheme simply treats each formation as a time bin. The lithostratigraphy was devised by *Plummer & Moore (1921)* and dated based on the marine strata which intercalate with the terrestrial strata. This provides a finer resolution than the land vertebrate faunachrons (11 bins rather than five) and later refinements of the lithostraigraphy (e.g., *Hentz, 1988*).

The fourth and final binning scheme uses a stochastic approach, in an attempt to address the time averaging that occurs when coarse time bins covering several million years are employed. The ages of the top and bottom of each formation were used as maximum and minimum bounds on the ages of each specimen known from within that formation. The period of time under study was split into half-million-year time bins, and each specimen was assigned at random to one of the bins between its maximum and minimum age brackets. One hundred such datasets were generated, and the analyses of diversity and extinction rate were applied to all 100. Such stochastic methods have been shown to provide more accurate estimates of standing diversity than binning approaches, even when the origination and extinction are biased towards coinciding with the boundaries of bins (*Gibert & Escarguel, 2017*).

For all four binning schemes, the absolute ages were derived from *Lucas (2017a)* and *Lucas (2017b)*, using his correlations of the formations to the international stages. Thus, the Littlecrotonian LVF and the San Angelo formation are deemed to be latest Kungurian rather than Roadian. For most of the binning schemes, this does not make a difference; when analysing only the Texas "best section", the question of whether an extinction event is identified between the Redtankian and Littlecrotonian is more relevant than the precise timing of the boundary. Where the absolute ages do make a difference is in calculating extinction rates using the stochastic binning scheme. By compressing the Redtankian and Littlecrotonian into a smaller period of time, the density of the specimens sampled is increased. This will lower extinction rates estimated under the gap-fillers method (see below): counts of two-timers will increase and counts of part-timers and gap-fillers will decrease (terminology from *Alroy, 2014*). The use of a Kungurian age for the Littlecrotonian is therefore more conservative, biasing against the inference of a mass extinction.

## Diversity and rate estimates

For each time bin in each binning scheme, diversity (species richness) estimates were calculated using two methods. The first is a taxic diversity estimate, a simple count of the number of species observed in each time bin without sampling correction. The second employs shareholder quorum subsampling (SQS; *Alroy, 2010a*; *Alroy, 2010b*), which standardises the coverage (the proportion of the rank abundance distribution sampled) in each time bin, and has been shown by both simulation studies and empirical data to be a robust method of correcting for preservation and sampling heterogeneity (*Alroy, 2010a*; *Alroy, 2010b*; *Chao & Jost, 2012*; *Close et al., 2018*). Coverage is measured using Good's U (the proportion of singletons relative to the total sample size). Diversity was estimates at four levels of coverage: 0.6–0.9 at intervals of 0.1 (a quorum of 0.6 allowed diversity to be calculated in all time bins in all binning schemes). SQS diversity estimates were calculated in R using version 3.3 of the function available on the website of John Alroy (http://bio.mq.edu.au/~jalroy/SQS.html). The stochastic binning method allows the implementation of the more precise and accurate methods of calculating extinction rates using the gap fillers method (*Alroy, 2014*). Since this method is based on estimating sampling from the patterns of occurrences in a moving "window" covering four time bins, it is impractical to apply it to the short time series produced by the three other binning strategies. The gap-fillers method was implemented, applying the "second for third" correction (*Alroy, 2015b*) to increase precision, using custom functions written in R. As suggested by *Alroy (2014)*, sampling heterogeneity was accounted for by classical rarefaction (standardising the sample size by number of occurrences) rather than by standardising coverage. Ten thousand subsampling iterations were carried out, each drawing five occurrences per time bin. Origination rates were calculated using the same equations; the methods used to calculate sampling apply equally well in reverse (*Alroy, 2014*). This was carried out using custom code, provided in Data S3.

# RESULTS

## Redefined land vertebrate faunachrons

When clustering the formations using CONISS, a number of changes are made to the boundaries of the LVFs (Fig. 1). The Littlecrotonian and Redtankian remain as they were defined previously. The lower boundary of the Mitchellcreekian is shifted downwards to include the Belle Plains Formation, found to cluster more closely with the Clyde than the Admiral Formation. The Admiral Formation itself clusters with the Putnam Formation, and so the Seymourian LVF is redefined to include these two. Thus, the Coyotean LVF contains only the Pueblo and Moran formations.

It is worth clarifying here that this analysis is not intended to cast doubt on the LVFs as originally defined; rather, they represent a biostratigraphic scheme more specific to the Texas section. The changes to the Coyotean LVF are most likely due to this being primarily defined by taxa from the well sampled early Cisularian localities in New Mexico (*Lucas, 1998*; *Lucas, 2017b*), which are not included in this study.

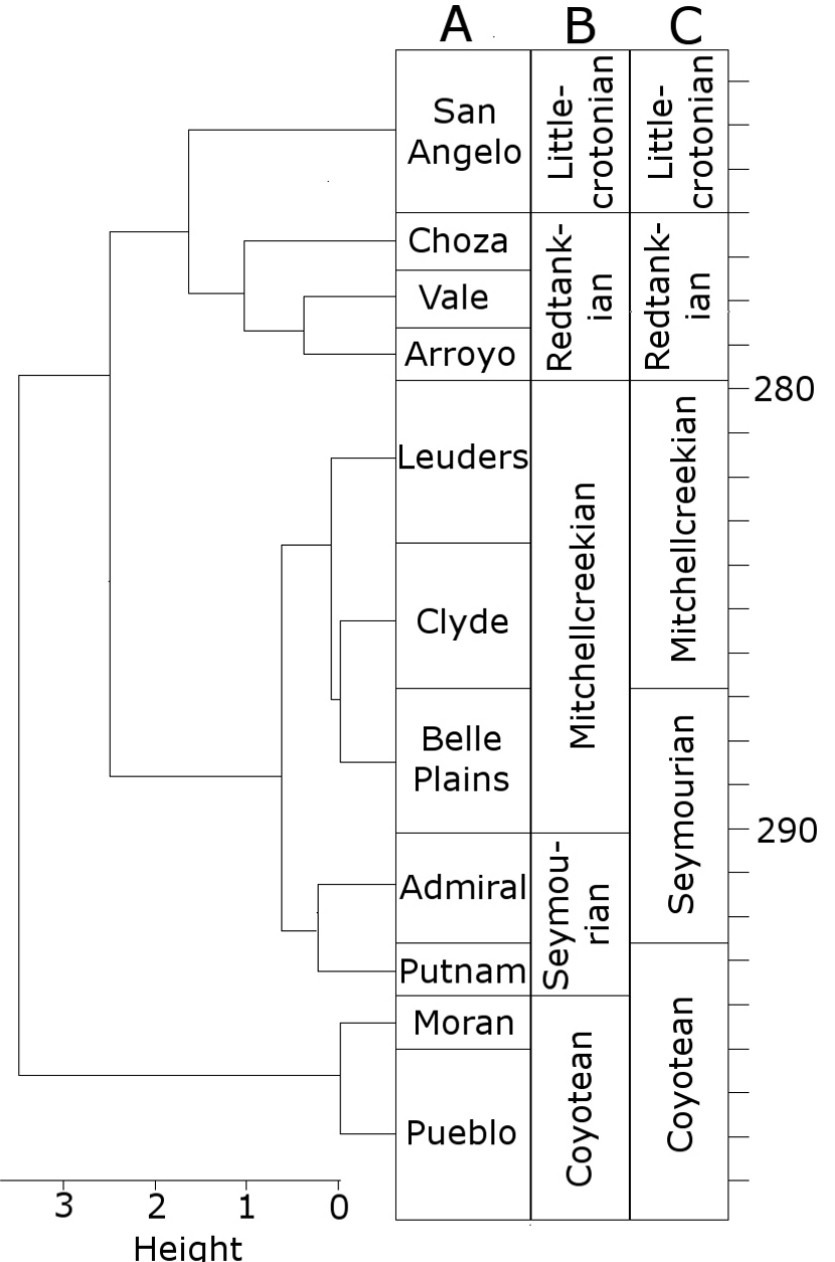

**Figure 1** **The time bins used in the diversity analysis.** (A) The cluster dendrogram indicating the grouping of the formations by CONISS; (B) the tetrapod bearing formations in Texas; (C) the Land Vertebrate Faunachrons (LVFs) redefined by CONISS; (D) the original LVFs.

## Diversity estimates

Raw, uncorrected species and genus-level diversity estimates indicate a substantial fall in diversity between the Redtankian and Littlecrotonian, based on all four time-binning schemes (Fig. 2). The finer resolution time bins (formation-level and half-million-year

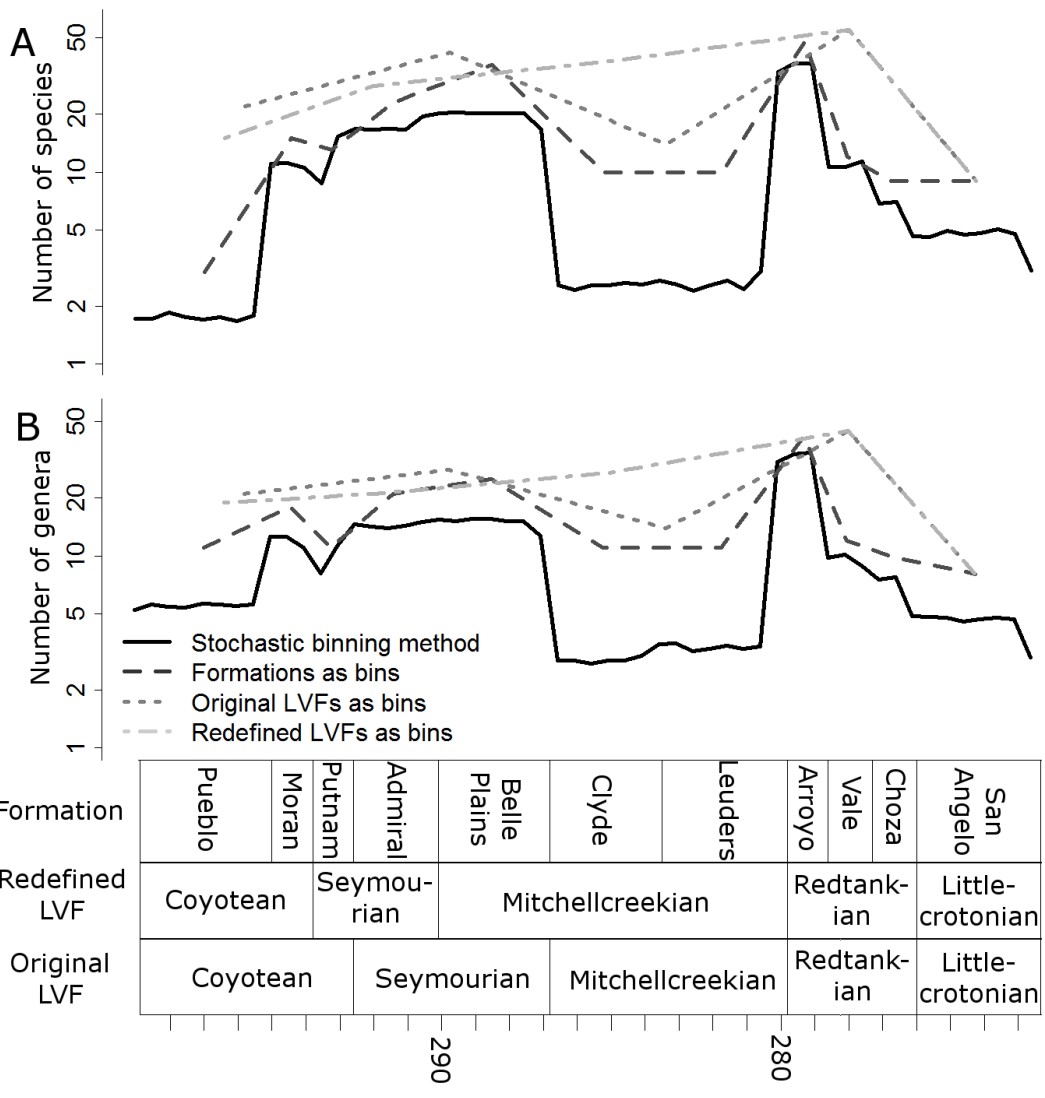

**Figure 2  Taxic diversity estimates.** Diversity estimates without correcting for sampling, using four different methods of time-binning the data. (A) Species level diversity estimate; (B) Genus level diversity estimate.

time bins) indicate that the Arroyo Formation represents the peak richness, and number of genera and species declined throughout the Redtankian.

When the data are binned by the Land Vertebrate Faunachrons (whether original or redefined), subsampling by SQS supports the Littlecrotonian as the time of lowest diversity (Figs. 3 and 4). The status of the Redtankian as a diversity peak is less clear; when the original LVFs are used, the Mitchellcreekian is found to contain a similar richness to the Redtankian (Fig. 3). However, the redefined LVFs indicate a substantial increase between these two bins (Fig. 4).

The higher-resolution-binning schemes both indicate the drop in subsampled diversity occurs throughout the Redtankain (Figs. 5 and 6). The Arroyo formation produced the

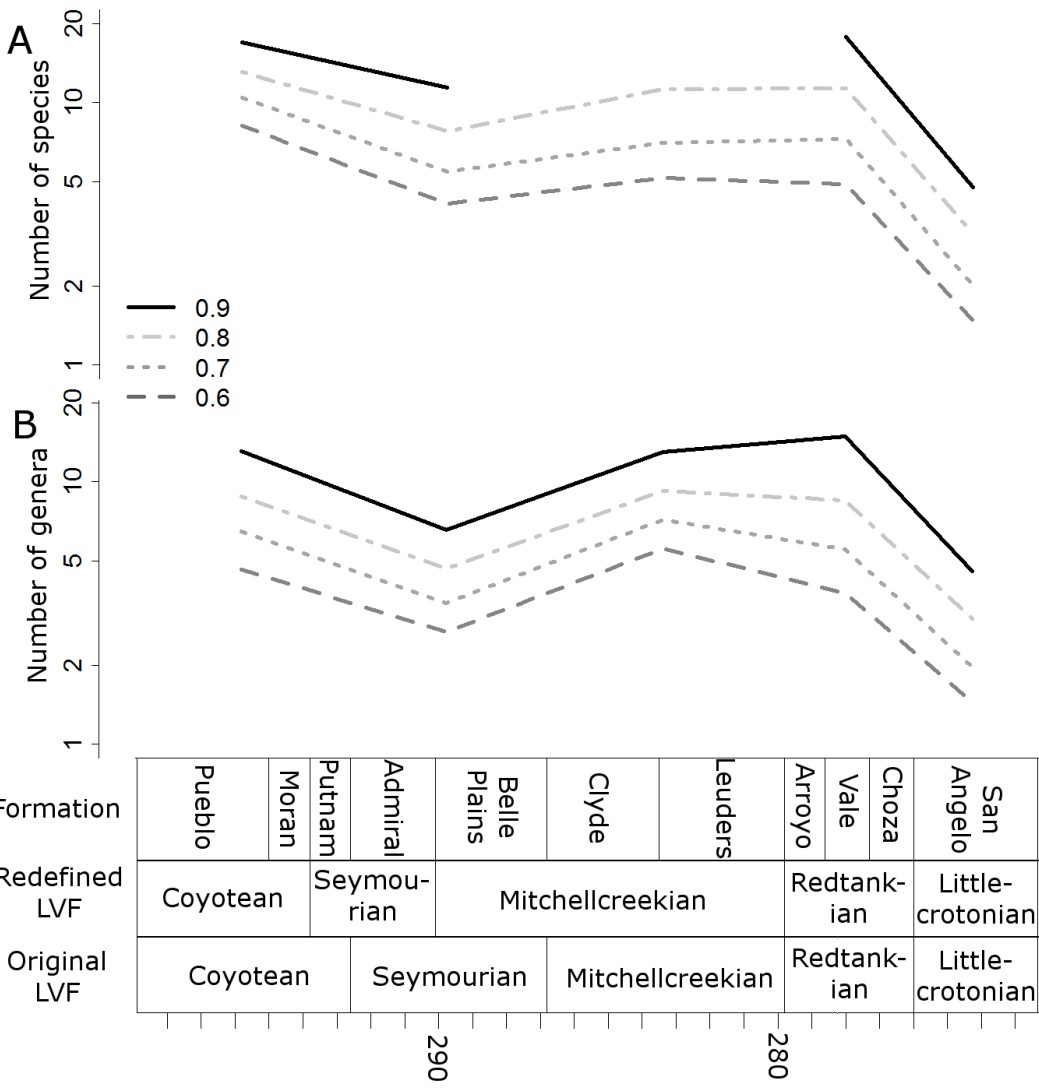

**Figure 3** **Subsampled diversity estimates (Original Land Vertebrate Faunachrons).** Numbers of species (A) and genera (B) in each land vertebrate faunachron (original definitions), corrected for sampling heterogeneity using shareholder quorum subsampling. Legend indicates quorum level.

highest species and genus richness of this faunachron, and the diversity deceases in the Vale Formation and reaches a trough in the Choza formation. When subsampling is applied, species and genus richness is found to increase slightly between the Choza and San Angelo formations.

## Extinction and origination rates

Three peaks in extinction rate are identified at the both at the genus and species level: at the top of the Belle Plains, Arroyo and Choza formations (the latter being the largest) (Fig. 7). During the time covered by the Vale Formation, extinction rates fall, but remain above background levels. The principal difference between the species and genus curves is

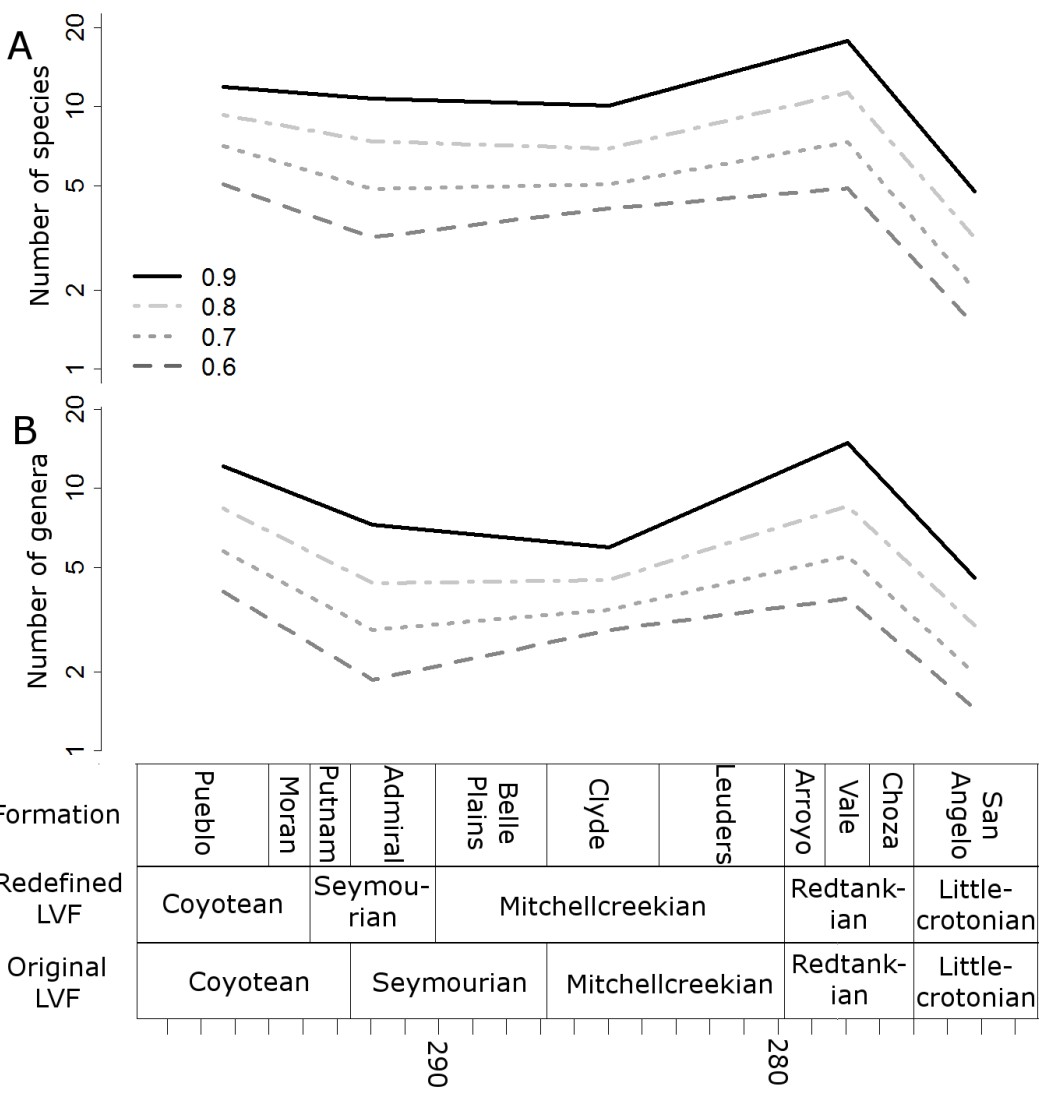

**Figure 4** **Subsampled diversity estimates (redefined Land Vertebrate Faunachrons).** Numbers of species (A) and genera (B) in each land vertebrate faunachron (definitions based on CONISS), corrected for sampling heterogeneity using shareholder quorum subsampling. Legend indicates quorum level.

the relative height of the Belle Plains extinction peak; at the species level it is higher than the Arroyo peak.

Peaks in origination rates at the species level are observed at the bottom of the Arroyo and the San Angelo formations (Fig. 8). The former of these peaks is not observed at the genus level, although the latter is still prominent.

## DISCUSSION

Having argued against an extinction of tetrapods across the Kungurian/Roadian boundary (due to the inappropriate time-binning strategies used in other diversity studies and the disagreement over the age of the San Angelo and Chickasha formations), *Lucas*

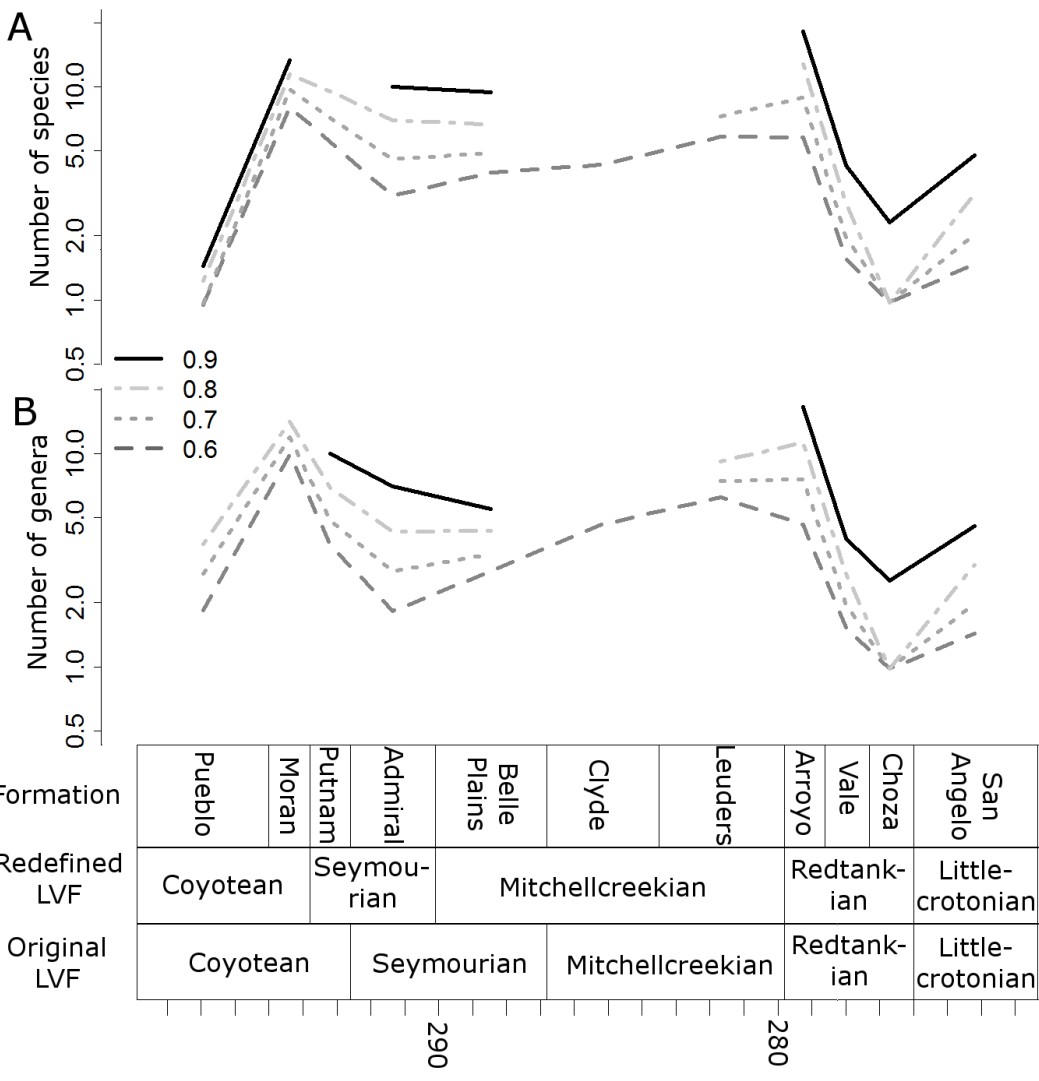

**Figure 5 Subsampled diversity estimates (formations).** Numbers of species (A) and genera (B) in each formation, corrected for sampling heterogeneity using shareholder quorum subsampling. Legend indicates quorum level.

*(2017a)* briefly examined the possibility of a mass extinction between the Redtankian and Littlecrotonian LVFs in the "best section" of Texas. Although he noted a peak in extinction rates during the Redtankian and a decrease in genus richness during the Littlecrotonian, he was dubious over the reality of a mass extinction. First, he suggested that families previously suggested to be major components of the extinction, Edaphosauridae and Ophiacodontidae (*Brocklehurst, Kammerer & Fröbisch, 2013*), had already disappeared prior to the end of the Redtankian. Lucas also examined diversity changes through the Redtankian using the specimen lists compiled by *Olson (1958)* and *Olson (1989)* for the Arroyo, Vale and Choza formations, demonstrating that diversity was decreasing throughout the Redtankian, rather than there being a single decline at the end of the LVF.

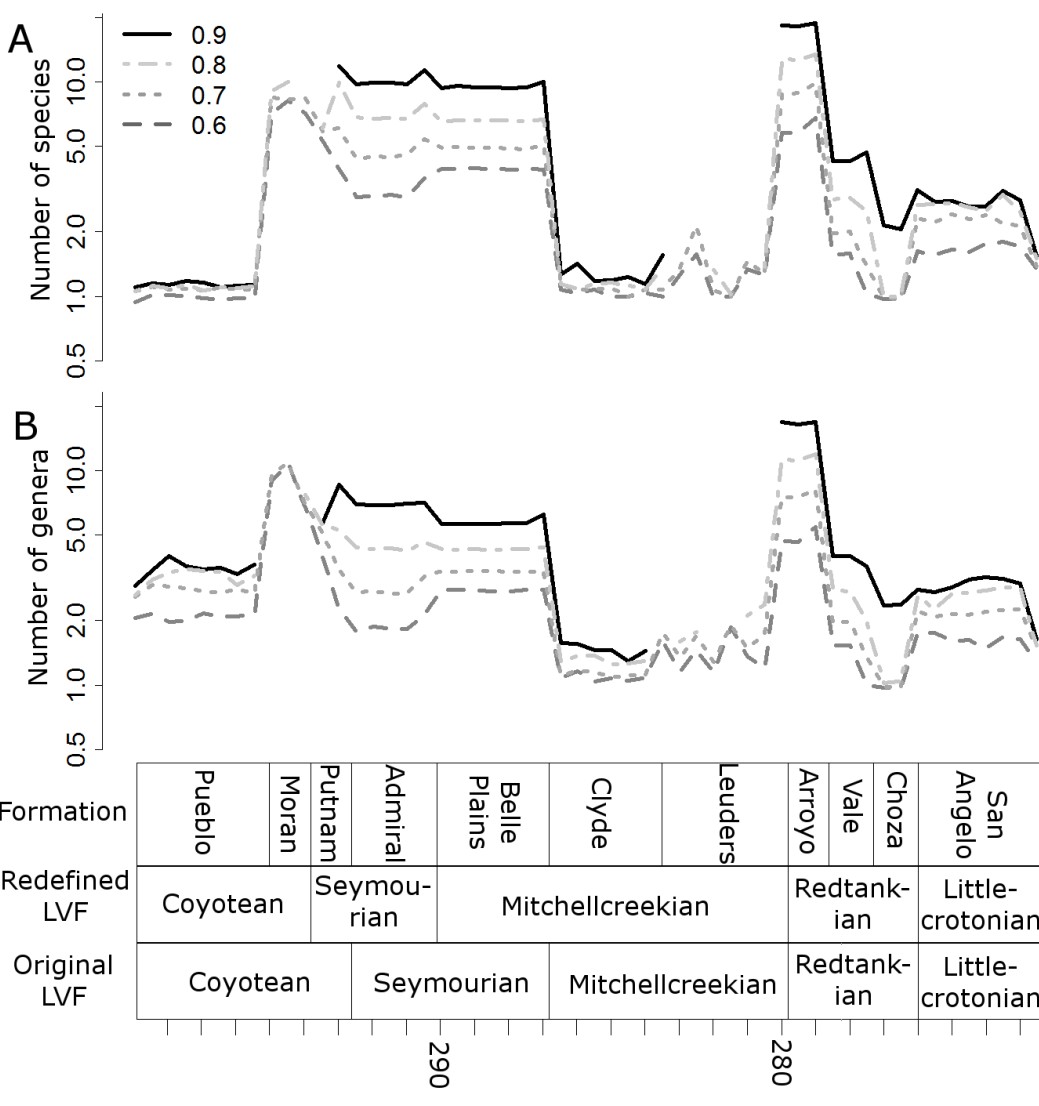

**Figure 6** **Subsampled diversity estimates (half-million-year time bins).** Means of the numbers of species (A) and genera (B) found in each half-million-year time bin in each of the 100 stochastic distributions of specimens, corrected for sampling heterogeneity using shareholder quorum subsampling. Legend indicates quorum level.

All diversity estimates presented here support a decrease in species and genus richness between the Redtankian and Littlecrotonian. The diversity estimates at finer stratigraphic scales support the observations of *Lucas (2017a)*: the decline occurs throughout the Redtankian from a peak in the Arroyo Formation to a trough in the Choza Formation, followed by a slight, but not substantial, recovery in the San Angelo Formation. The same inferences may be made from origination and extinction rates. While origination rates peak at the bottom of the Arroyo Formation (explaining the peak in species richness at this time), extinction rates are noticeably higher in the Arroyo Formation than the background rates experienced for most of the early Permian. Only once prior to this are extinction
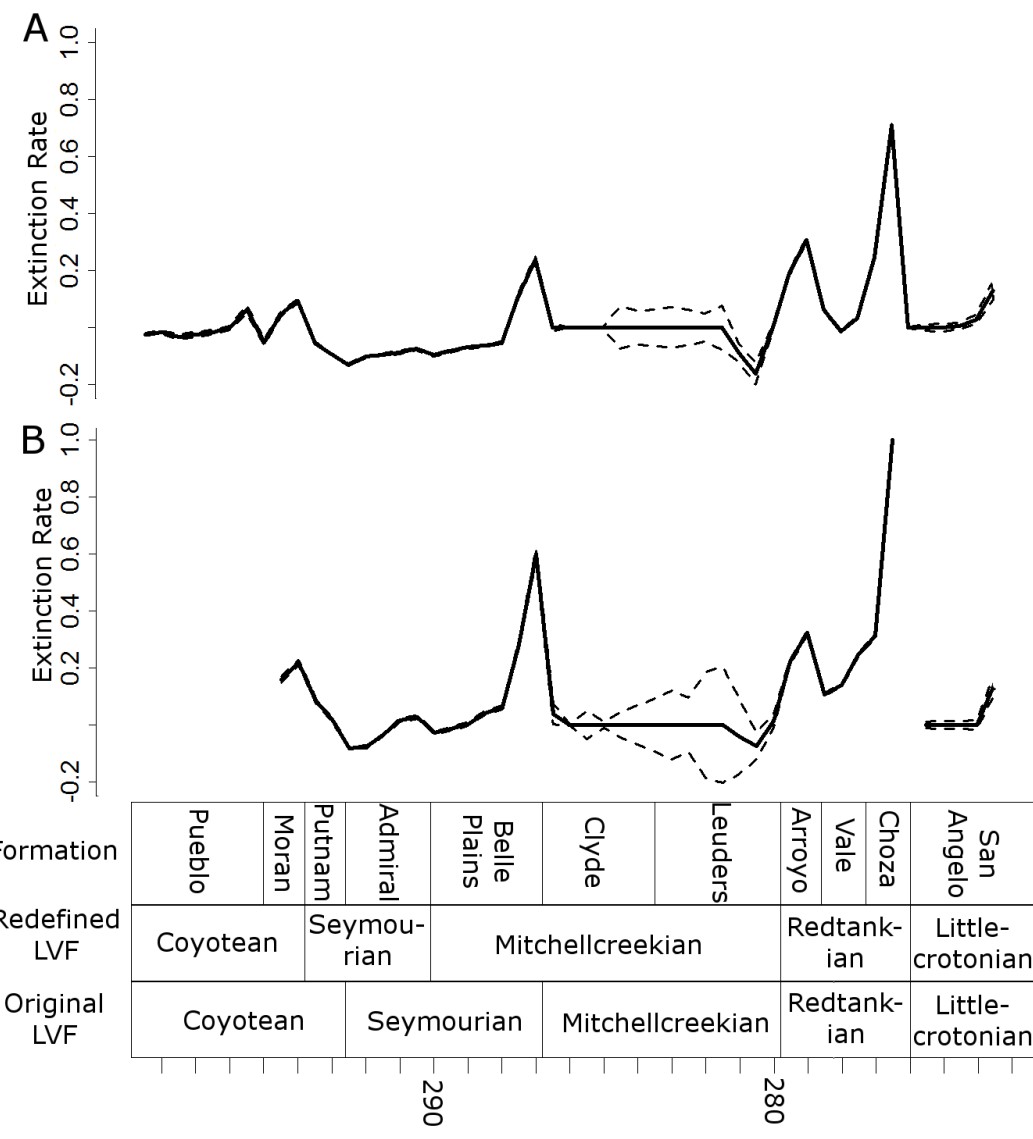

**Figure 7  Extinction rates.** Median (thick black lines) of the extinction rates calculated for each half-million-year time bin in each of the 100 stochastic distributions of specimens at the genus (A) and species (B) levels. Dashed lines indicate standard error around the median.

rates reliably inferred to reach similar levels: at the top of the Belle Plains Formation. The extinction rates experienced in the Choza Formation are considerably higher than any other time in the early Permian, and origination rates do not rise until later, at the bottom of the San Angelo Formation (coinciding with the post-extinction recovery).

Does this period of elevated extinction rates and declining diversity constitute a mass extinction? *Lucas (2017a)* argued not, since it was a prolonged decline throughout the Redtankian LVF. Unfortunately, there is no set definition of a "mass extinction", and while the general consensus does seem to be elevated extinction over a short period of time, there is no indication of how short a time that should be. Discussion of mass extinctions in the

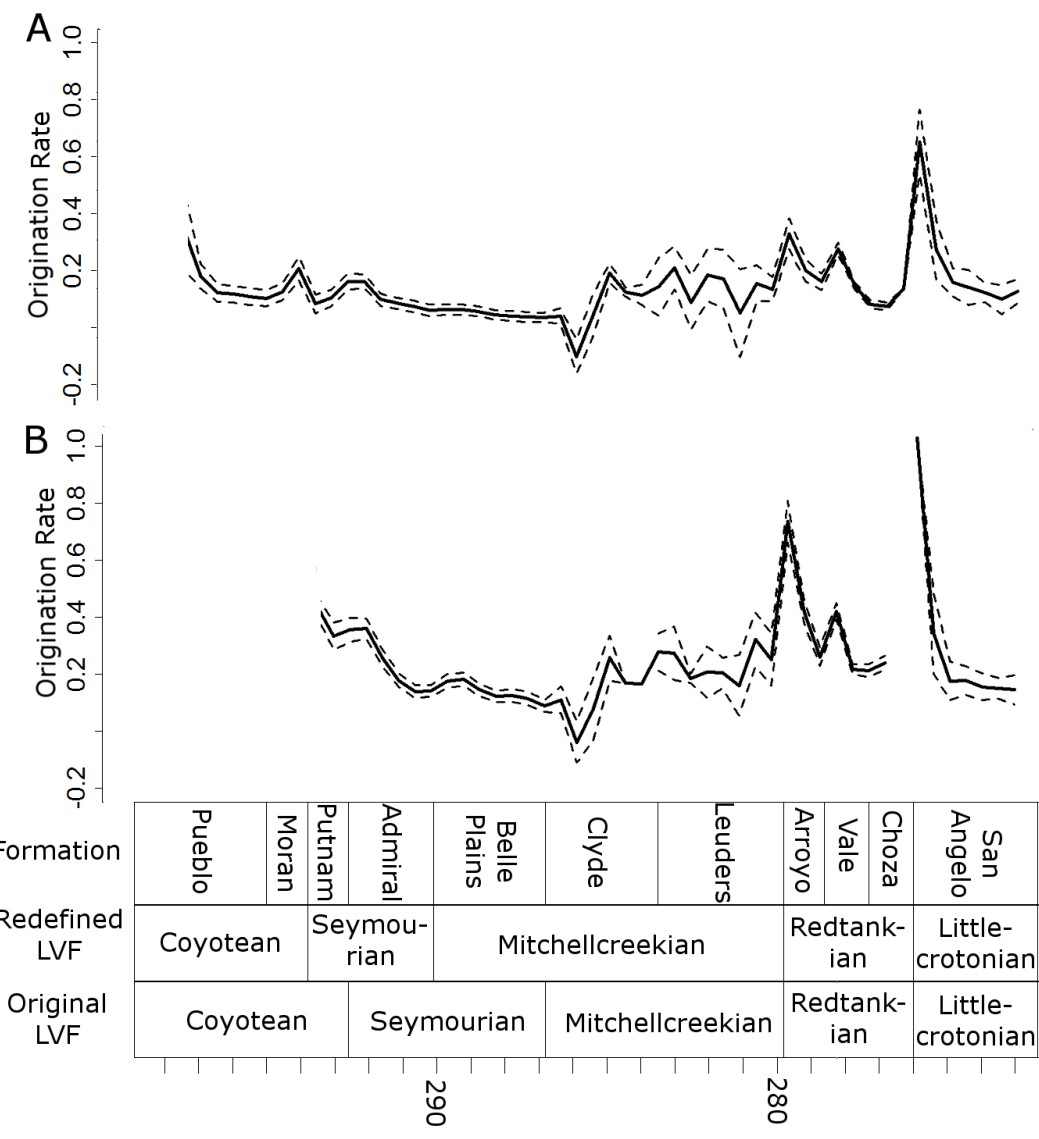

**Figure 8  Origination rates.** Median (thick black lines) of the origination rates calculated for each half-million-year time bin in each of the 100 stochastic distributions of specimens at the genus (A) and species (B) levels. Dashed lines indicate standard error around the median.

scientific literature have included events where extinction rates were substantially higher than background rates over periods of millions of years. For example, discussion of the late Devonian mass extinction (one of the "big five" mass extinctions) has in the past suggested a duration of up to three million years (*Racki, 2005*); the end Triassic extinction (another of the big five) is thought to represent periods of elevated extinction rate bracketing the entire Rhaetian stage (*Ward et al., 2001*; *Ward et al., 2004*), a duration of almost seven million years based on the most recent timescale of the International Commission on Stratigraphy. Moreover, if one is to follow the stratigraphic ages espoused by *Lucas (1998)*, *Lucas (2002)*, *Lucas (2004)*, *Lucas (2006)*, *Lucas (2017a)* and *Lucas (2017b)*, the Redtankian

would be compressed into a period covering less than four million years. During these four million years, extinction rates remain consistently higher than background levels. The Arroyo Formation records a substantial increase in extinction, and the Choza Formation records extinction rates that have more-than doubled those of the Arroyo, higher than in any other formation. The number of tetrapod species observed in the Choza Formation is less than a quarter of those observed in the Arroyo Formation, and subsampling does not diminish the extent of the diversity loss.

It is worth noting at this point that mass extinctions appearing in the fossil record as prolonged declines is an issue that has a long history of discussion in the published literature, going back to the work of *Signor & Lipps (1982)*. The fact that the last appearance of a taxon in the fossil record is not its last true appearance, combined with differential preservation probabilities of different taxa, causes a set of species, which in reality died out nearly simultaneously, to appear to have died out over a longer period of time (*Butterfield, 1995*), a phenomenon dubbed the Signor-Lipps effect. *Lucas (2017a)* acknowledged the Signor-Lipps effect in his introduction but did not mention it in his discussion of specific extinction events. He also employed no sampling correction when examining diversity and extinction rate, instead arguing that "whatever biases exist may be roughly equivalent in the Permian tetrapod record across times and localities" (*Lucas, 2017a*, p. 35). This is simply not true: there is a wealth of literature detailing analyses of the quality of the fossil record of Paleozoic tetrapods, all suggesting the opposite and emphasising the need for sampling correction (*Benson & Upchurch, 2013*; *Brocklehurst, Kammerer & Fröbisch, 2013*; *Brocklehurst & Fröbisch, 2014*; *Brocklehurst et al., 2017*; *Verrière, Brocklehurst & Fröbisch, 2017*).

Another argument put forward by *Lucas (2017a)* to show that Olson's Extinction does not qualify as a genuine mass extinction is that many of the clades previously deemed to have died out at this time actually disappeared before the end of the Redtankian, and the number of actual casualties of the event, at the family level, was very restricted. *Brocklehurst, Kammerer & Fröbisch (2013)* previously noted Edaphosauridae and Ophiacodontidae as "casualties", but *Lucas (2017a)* countered that the former's last appearance is from the Arroyo Formation rather than the end of the Redtankian LVF, and that the latter is not known from beyond the Mitchellcreekian LVF. In the case of the Ophiacodontidae, this is actually not the case, and the family survived into the Redtankian. *Lucas (2017a)* based his assertion on the last record of *Ophiacodon*, and the abundant record of *Varanosaurus*, represented in the Arroyo Formation by the species *V. acutirostris* (*Broili, 1904*; *Case, 1907*; *Case, 1910*; *Romer & Price, 1940*) and *V. witchitaensis* (N Brocklehurst, pers. obs., 2013), was discounted as representing a taxon of uncertain assignment. However, almost three decades of study, both anatomical and cladistic, support the ophiacodontid affinity of *Varanosaurus* (*Sumida, 1989*; *Berman et al., 1995*; *Benson, 2012*; *Brocklehurst et al., 2016*), and I see no reason not to count it as the youngest record of Ophiacodontidae. Regarding Edaphosauridae, only one species of *Edaphosaurus* is known from the Arroyo formation (*E. pogonias*), but it still represents one of the most abundant herbivores in this fauna (Data S1). Neural spine material of *Edaphosaurus* is also known from the Hennessey Formation (*Daly, 1973*), a Redtankian aged formation in Oklahoma. It is clear, therefore, that both

Ophiacodontidae and Edaphosauridae survived into the Redtankian. While they may not have survived beyond the lowest of the Redtankian formations, this does not remove them from the Olson's Extinction casualty list. As already discussed, extinction rates were raised considerably above background levels throughout the Redtankian, and extinctions of the taxa of the Arroyo formation should be included in event.

Even if we are to limit our discussion to clades which went extinct at the end of the Choza Formation, there are still multiple clades above the genus level which may be included in the list of casualties of Olson's extinction, mostly amphibians. Probably the most prominent are the Eryopidae, since they represent one of the few cases where we have data on their disappearance from both palaeoequatorial (USA) and palaeotemperate localities (*Brocklehurst et al., 2017*). Eryopids represent among the most abundant of the large amphibians throughout the Cisuralian, and *Eryops* itself survives until the Choza Formation (Data S1). Crucially, two eryopid species are known from the latest Kungurian of Russia: *Clamorosaurus borealis* and *C. nocturnus* from the Inta Formation (*Gubin, 1983*). Eryopids are not known beyond the Kungurian in either the palaeoequatorial or palaeotemperate latitiudes beyond this time (*Brocklehurst et al., 2017*). The Trimerorhachidae and Lysorophia are two more clades highly abundant throughout the Cisuralian, but which do not survive beyond the Choza Formation (Data S1). Both are also known from the Redtankian aged Hennessey Formation in Oklahoma, but not from the Littlecrotonion Chickasha Formation (*Brocklehurst et al., 2017*). The Choza Formation represents the greatest peak in extinction rate in the entire Cisuralian in this particular section, both at the genus and species level, with extinction rates more than double the next highest peak. Therefore, even if one discounts the losses occurring earlier in the Redtankian, it is difficult to deny the presence of a severe extinction event at the Redtankian/Littlecrotonian boundary.

There has not been much work on the environmental context surrounding this event, but the extended period of extinction has been suggested to coincide with substantial climatic changes recorded in the Texas sequence. The sediments of the Vale Formation record a transition from an environment dominated by large perennial streams to one of braided channels, with indications that flow was interrupted by substantial periods of drying (*Olson, 1958*). The Choza Formation indicates a trend towards ever increasing aridity, with the uppermost deposits formed almost entirely from anhydrites (*Olson, 1958*). More work needs to be done on this crucial time period, and until further research on environmental changes at this time is carried out these questions cannot be answered with great certainty.

## CONCLUSIONS

No matter what time-binning scheme is employed, no matter whether the data is examined at the species or genus level, and no matter whether the data is corrected for sampling or not, a substantial mass extinction event is observed in tetrapods during the Redtankian Land Vertebrate Fanuachron. Throughout the Redtankian, extinction rates were raised substantially above background levels, rising to a peak in the uppermost Choza Formation.

Tetrapod diversity declines throughout this period, and by the end of the Redtankian, species richness is less than a quarter of that observed at the start.

## ACKNOWLEDGEMENTS

I would like to thank Roger Benson and John Alroy for helpful comments. The reviewers Kenneth Angielczyk and Bruce Rubidge and editor Graciela Piñeiro provided many helpful suggestions that greatly improved the quality of the paper. I am also grateful to collection managers who gave access to their collection catalogues and the collections themselves: Jennifer Larsen (Sam Noble Oklahoma Museum of Natural History), William F. Simpson (FMNH), Dan Brinkman (Peabody Museum of Natural History), Jessica Cundiff (MCZ), Carl Mehling (AMNH) and Pat Holroyd (UCMP). I would also like to thank Jörg Fröbisch and Deutsche Forschungsgemeinschaft.

### Funding

The author did not receive funding for this study.

### Competing Interests

The authors declare there are no competing interests.

### Author Contributions

- Neil Brocklehurst conceived and designed the experiments, performed the experiments, analyzed the data, prepared figures and/or tables, authored or reviewed drafts of the paper, approved the final draft.

### Data Availability

The raw data and code are provided in the Supplemental Files.

### Supplemental Information

Supplemental information for this article can be found online at http://dx.doi.org/10.7717/peerj.4767#supplemental-information.

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
