# Peer review of "An examination of the impact of Olson’s extinction on tetrapods from Texas"

_PeerJ, doi:10.7717/peerj.4767_

## Round 0.1 · original submission · Minor Revisions

Dear author,

I am glad to announce that we have now two review reports about your manuscript, and both reviewers encountered it very interesting and recommended its publication in PeerJ.

Some of the main reviewers concerns and comments point out to a necessity of reinforcing your results to make the Olson´s extinction event more reliable. Other important observation among others that the reviewers marked in the attached annotated PDF, is the necessity to know how you arrived to the absolute ages that support your conclusions. I recommend that you address all of them carefully.

On the same way, I think that to strength your results you can offer crucial data that otherwise would be subject of curiosity by the readers:

1) All the specimens that you used in your study posses unambiguous stratigraphic provenance information? And what about the lithology? Do you think that you can provide at least minimal descriptions about the lithological and thus environmental changes observed through the levels where the specimens were collected?
2) There could be a taphonomic bias influencing the abundance and diversity of the discoveries?
3) Have you controlled the inherence of eventual time averaging processes?
4) Even considering that the extinction existed, could you link it to some particular catastrophic event that caused such declination in the tetrapod diversity between the Redtankian and the Littlecrotonian?

I hope that you find these suggestions useful to improve your manuscript and look forward to see the revised version soon.

Kind regards,
Graciela Piñeiro

·

Basic reporting

General comments: This manuscript investigates evidence for “Olson's extinction” and apparent decline in tetrapod diversity near the end of the early Permian. The reality of this extinction has been debated for some time based on potential problems associated with differences in geographic sampling, binning of samples, and correlations among different rock units (and with the SGCS). This paper attempts to deal with these issues by focusing on diversity patterns from just one geographic area (the classic Permian redbeds of Texas), utilizing a specimen-level dataset. In many ways it is a follow-on and response to Lucas' (2017) paper in Earth-Science Reviews, which looked at a similar question but using a less detailed data set and without attempting to control for sampling effects. The result here suggest that there is indeed a decline in tetrapod diversity, and an increase in extinction rates, at this time, regardless of how samples are binned temporally and whether sampling is controlled for. I found the paper to be very clear and well-written, and I think the analyses are convincing. I have a few comments and questions below, but I think most of them should be fairly easy to address. Of these, I think the most important are 1) why the older Plummer and Moor lithostratigraphy is used instead of the more recent (and arguably more directly applicable to the specific rocks in question) revision of Hentz (1988); and 2) how the seemingly very precise numerical ages were assigned to the beginnings and ends of the LVFs. The author says that the dates were taken from Lucas (2017), but as far as I can tell he doesn't assign numerical ages to the the boundaries in either his Earth-Sciences Reviews paper or his Geological Society of London book chapter. Overall, I think the paper should be acceptable with minor revisions.

Experimental design

see comments to author

Validity of the findings

see comments to author

Additional comments

Abstract first sentence: ...is not actually a sentence.

Line 31: whereas instead of while

Line 35: I don't know if I would characterize the transition Olson described as resulting in more trophic levels. It's more of a transition in the main path of energy flow in the terrestrial component of the ecosystems, with terrestrial productivity being passed more directly to terrestrial secondary consumers via tetrapod (and/or insect) herbivores.

Line 57: whereas instead of while

Line 59: The shift in geographic areas of fossil preservation going from the early to middle Permian doesn't seem like the key issue in figuring out the timescale of the event. Instead, the lack of radiometric dates for key formations on either side of the event is the most significant hindrance.

Line 74: remove the before extinctions

line 75: change they to he

Line 77: I recommend starting a new paragraph here. The discussion of the debate over the age of the formations seems distinct enough to warrant it.

Line 112: previous instead of previously

Line 122: General nitpicky semantic comment: data are plural, so I recommend chaning your verbs to reflect this throughout the manuscript

line 124: change on to in

Line 133: Because your dataset includes numbers of specimens, it can address how much of a problem Lucas' concern with singletons might be. Therefore, I think it would be useful to provide a couple of summary statistics about this (e.g., percentage of taxa in your dataset that are known from just one specimen, percentage that just occur in one stratigraphic or biochronological unit), probably at the start of the results.

Line 140: It might be worth noting that they are biostratigraphic bins that are essentially based on the section in which you're working. As such, they likely capture details that might be obscured in the global timescale, which is patched together with information from various geographic areas and environments.

Line 152: I'm not sure if you provide code for the analysis in your in press Evolution paper. If you don't, you should consider doing so here (and you may want to even if you do include code in the Evolution paper).

Line 158: These lithostratigraphic units certainly provide a good connection with historical research in the area. However, Hentz (1988) proposed a revised lithostratigraphy that applied more directly to the terrestrial strata. Is there a reason that you didn't use that? Even if it is something as simple as Hentz's scheme not providing many more bins than the LVF binning scheme, you should note this.

Line 168: A few comments here. First, the paper you cite for the ages is Lucas' Earth-Science Review paper from 2017, but that paper doesn't delve into the details of how the correlations between his LVFs and the SGCS were determined. Instead he refers readers to his 2017 Geological Society of London book chapter, which does provide the details. Since the source of the ages is critical to your inferences about rates, I think it would be good to cite this paper as well. Second, the it is perhaps worth noting that Lucas' ages for the Texas LVFs are mostly based on (marine) biostratigraphy and radiometric ages from mostly geographically distant areas. Third, Lucas doesn't specify exact numerical dates for the start and end of his different LVFs, so it would be useful to describe how you determined the seemingly high-precision dates that you present in the supplement, and which are presumably bound up in the rates that you calculated.

Line 187: estimated instead of estimates

Line 196: again, consider presenting code in the supplement

Fig. 1: The lettered labeling for the parts of this figure seem awkward. I don't think the dendrogram needs a label, and I would change the labels on the time scale to SGCS, this study, and Lucas (2017), respectively.

Line 208: Do you think the change in the placement of the boundary of the Coyotean here stems at all from the fact that the LVF is defined based on a fauna from New Mexico, whereas the others in this sequence are defined based on faunas from Texas?

Line 212: I think you mis-characterize the results in Fig. 2 a bit here. Yes, there is a substantial drop between the Redtankain and Littlecrotonian when data are binned at the LVF scale, but your finer binning schemes seem to indicate that most of that drop was in the Redtankian, and potentially quite early in the Redtankian, such that the observed low diversity of the Redtankian is more a continuation of an established parttern than something new.

Line 232: there seems to be something missing from this sentence. I assume you mean something like: “Having argued against an extinction of tetrapods at the Kungurian/Roadian boundary (due to...”

General comment on the discussion: Is it worth considering how origination rates changed across this time as well? Lucas (2017) talks a little bit about this, and suggests that origination rates were somewhat elevated in the Redtankian and Littlecrotonian. It would be interesting to see what the patterns look like given your revised/finer scale binning. Your argument of a mass extinction would also be further reinforced if you could show a decrease in origination rates corresponding with an increase in extinction rate.

Line 331: you should note here that this is observed in the Texas section

Ken Angielczyk

·

Basic reporting

REVIEW OF :
An examination of the Impact of Olson’s extinction on Tetrapods from Texas – by N Brocklehurst

The matter of the validity and impact of Olson’s extinction on early- middle Permian tetrapod faunas has been the subject of much debate in the scientific literature but there has not been clear resolution largely because of the geographically disparate nature of the sedimentary successions hosting relevant fossil faunas. The subject is important as this is the first large extinction event affecting tetrapod faunas in the continental realm.

This paper examines the tetrapod fossil record from different formations of the Texas sedimentary succession, arguably the temporally most complete Early Permian succession, at a finer stratigraphic level than that used in previous analyses. Species richness in the different formations has been determined using four different binning schemes. The author convincingly makes a case that extinction rates are greater than background rates, but the peak occurs in the Choza Formation. He concludes that Olson Extinction is a real event and is not an artefact of binning methodology. In my opinion this is a useful paper and should be published.

Basic reporting

The paper is well written with good language usage, and the structure conforms to PeerJ standards. The abstract is concise, and the introduction provides a comprehensive literature review setting out the problem which is being addressed. The Results and Discussion are appropriately presented. I suggest that the Conclusion needs to be expanded to set out what has been achieved by this paper.
Literature is well referenced, and the figures are all relevant. The raw data is adequately presented in two tables providing number of specimens of particular genera and species from individual formations.

Experimental design

The project is not dependant on experimentation. The methodology is well set out and explained and the research question well defined and relevant. The paper sets out the intention to approach the question of the “existence” of Olson’s extinction from a different angel to that previously used by employing a binning strategy of the tetrapod genera and species from different formations in the fossil record from the Texas stratigraphic sequence. Four methods are used to define time bins which are well explained and carried out. The methodology is well set up and the raw data adequately presented as two tables in subsidiary information. In my opinion the methods are described with sufficient detail to replicate.

Validity of the findings

CONISS clustering of the formations suggest that the Mitchellcreekian be shifted downwards to include the Belle Plains Formation.

By employing the methodology explained using the four time-binning schemes, in the results section the paper convincingly indicates a substantial drop in diversity between the Redtankian and Little Crotonian and shows the peak of taxonomic richness to be in the in the Arroyo Formation.

The paper further points out three peaks in extinction fate (both at genus and species level) at the top of the Belle Plains, Arroyo and Choza formations.
The paper convincingly points out that a substantial tetrapod mass extinction event occurred in the Redtankian and as such this information deserves publication in PeerJ.

The conclusion does not adequately summarise the findings of the paper and should be fleshed out.

Additional comments

I have pointed out various relatively minor editorial modifications with annotations in the attached pdf. The following are specific points which need attention.
L26 – add Permian as a key word.
L 79 – include comas
L 82 – and other places pointed out in the text (Lines 117, 205, 213, 228 ). In formal stratigraphic nomenclature Formation (in this case Blaine Formation) requires an upper case “ F”. if it is used in the plural eg. San Angelo and Chickasha formations then it is lower case ‘f”
L84 – Riesz should be Reisz
L 112 previously studies – shodul be previous studies

---

## Round 0.2 · Minor Revisions

Dear author,

I have to recognize that you made important efforts to improve the manuscript by addressing the suggestions from the reviewers, but there are some matter that still remain doubtful for me (I could not ascertain them from your answers in the rebuttal letter). For instance, my main concern is the taphonomic bias, because you had a decreasing in diversity more pronounced during the Redtankian, and it should be well detected in the Vale Formation, right?. However, this unit possesses a good record of fossils, including well preserved, although not so much diverse flora and apparently does not show any sign of droughts. So, I still wonder what could cause the extinction, and if you have in fact more than one event represented. This is important because it could suggest that the extinction was resolved just in the latest Redtankian. These issues, joined to the high rate of taxa that are represented by just an individual, make necessary an interpretation. If preservation of the fossils is not favorable in the studied formations, you could have a lack of information that would affect your results, so, how your methodology will deal with this problem? You should discuss this more extensively in the manuscript. If you have reworked specimens, this issue can be affecting your results, unless you detect them and consider if they are useful for this kind of studies.

Well, I hope you can address these questions; maybe they seem more relevant to me because I do not know the stratigraphic section studied, although I tried to learn about the geological setting of the involved formations.

Best regards,
Graciela Piñeiro

---

## Round 0.3 · accepted · Accept

Dear author,

Since you are admitting that “More work needs to be done on this crucial time period, and until further research on environmental changes at this time is carried out these questions cannot be answered with great certainty”, coupled to what I could know from the bibliography (I never seen the stratigraphic succession that you are studying but I wrongly assumed that you did), it is logical that I ask for you to include a causality to the events that the program that you used is suggesting. Thus, I think that this issue and taphonomy, are crucial to validate your results. That was the reason for what I suggested that you include a paragraph about the events that support the proposed extinction, which, I repeat, seems to be enough proved. However, your work would be less creditable if you avoid treating the main constraints that your results can have: causality and taphonomic traits that could add weakness to the program interpretations. Thus, it is good that you have reinforced your hypothesis with all you have at hand, showing that you paid attention to these constraints (records based in just one, and fragmentary specimen, reworked specimens (which indeed would have been present if they come from conglomerates), time averaging, etc., which are apparently well solved by the methodology used (and so, you cited other contributions that support your statements), but being cautious with the causes, which are not yet well established. That is what I needed to see reflected in your manuscript to accept it for publication in PeerJ.

Sincerely,
Graciela Piñeiro

#